# Topical Chitosan-Based Thermo-Responsive Scaffold Provides Dexketoprofen Trometamol Controlled Release for 24 h Use

**DOI:** 10.3390/pharmaceutics13122100

**Published:** 2021-12-06

**Authors:** Luis Castillo-Henríquez, Pablo Sanabria-Espinoza, Brayan Murillo-Castillo, Gabriela Montes de Oca-Vásquez, Diego Batista-Menezes, Briner Calvo-Guzmán, Nils Ramírez-Arguedas, José Vega-Baudrit

**Affiliations:** 1National Laboratory of Nanotechnology (LANOTEC), National Center for High Technology (CeNAT), San José 1174-1200, Costa Rica; luis.castillohenriquez@ucr.ac.cr (L.C.-H.); gmontesdeoca@cenat.ac.cr (G.M.d.O.-V.); dbatista@cenat.ac.cr (D.B.-M.); 2Laboratory of Physical Chemistry, Faculty of Pharmacy, University of Costa Rica, San José 11501-2060, Costa Rica; pablo.sanabriaespinoza@ucr.ac.cr (P.S.-E.); brayan.murillocastillo@ucr.ac.cr (B.M.-C.); nils.ramirez@ucr.ac.cr (N.R.-A.); 3Laboratorios Infarma LTDA, Cartago 30801, Costa Rica; investigacionydesarrollo@laboratorioinfarma.com; 4Laboratory of Biopharmacy and Pharmacokinetics (LABIOFAR), Institute of Pharmaceutical Research (INIFAR), San José 11501-2060, Costa Rica; 5Laboratory of Polymers (POLIUNA), Chemistry School, National University of Costa Rica, Heredia 86-3000, Costa Rica

**Keywords:** chitosan, dexketoprofen trometamol, drug delivery, gelatin, NSAIDs, personalized medicine, smart polymers

## Abstract

Chronic and non-healing wounds demand personalized and more effective therapies for treating complications and improving patient compliance. Concerning that, this work aims to develop a suitable chitosan-based thermo-responsive scaffold to provide 24 h controlled release of Dexketoprofen trometamol (DKT). Three formulation prototypes were developed using chitosan (F1), 2:1 chitosan: PVA (F2), and 1:1 chitosan:gelatin (F3). Compatibility tests were done by DSC, TG, and FT-IR. SEM was employed to examine the morphology of the surface and inner layers from the scaffolds. In vitro release studies were performed at 32 °C and 38 °C, and the profiles were later adjusted to different kinetic models for the best formulation. F3 showed the most controlled release of DKT at 32 °C for 24 h (77.75 ± 2.72%) and reduced the burst release in the initial 6 h (40.18 ± 1.00%). The formulation exhibited a lower critical solution temperature (LCST) at 34.96 °C, and due to this phase transition, an increased release was observed at 38 °C (88.52 ± 2.07% at 12 h). The release profile for this formulation fits with Hixson–Crowell and Korsmeyer–Peppas kinetic models at both temperatures. Therefore, the developed scaffold for DKT delivery performs adequate controlled release, thereby; it can potentially overcome adherence issues and complications in wound healing applications.

## 1. Introduction

Chronic and non-healing wounds caused by different diseases (e.g., diabetes) demand special therapies for treating chronic inflammatory processes, infections, and poor tissue regeneration [1]. Despite current interventions, there is a growing demand for personalized and more effective treatments for healing wounds that improve drug delivery, life quality, and patient compliance [2].

Chitosan-based scaffolds are excellent options for topical drug delivery, at the same time they provide physical support for enhancing tissue regeneration and cell growth [3,4]. The amphiphilic nature of the polymer and the ease of tuning its physicochemical properties, such as the critical solution temperature (i.e., the temperature at which exists a balance in the competition between hydrophilic and hydrophobic polymer chains) make it a promising material for thermo-responsive drug delivery in wound healing [5,6,7]. In general, when modified, it exhibits a lower critical solution temperature (LCST) around 32–42 °C. These thermotropic phase transitions can be induced by proton transfer from the material to another cooperatively binded compound (e.g., β-glycerophosphate), or by the cross-linking with another polymer (e.g., poly(*N*-isopropylacrylamide)) [8,9,10]. This feature has been taken advantage of by cross-linking with polyvinyl alcohol (PVA) and gelatin, which improve thermal and pH stability, processability, and can also provide better-controlled release of drugs from the scaffold system [11,12].

An important attribute of topical chitosan-based scaffolds is the possibility of loading non-steroidal anti-inflammatory drugs(NSAIDs), like Dexketoprofen (DK), used as the tromethamine salt derivate, Dexketoprofen trometamol (DKT). The latter is a class 1 drug (i.e., high solubility and high permeability) according to the Biopharmaceutical Classification System (BCS) [13,14]. This non-selective cyclooxygenase enzyme (COX) inhibitor is the (S)-(+)-enantiomer of ketoprofen, and is used for the rapid management of mild to moderate pain with a maximum daily dose of 75 mg, or 25 mg every 8 h [14]. In addition, the drug is more potent, causes fewer side effects, and is more tolerable than the racemic compound [15].

Different approaches to the delivery of DKT have been made, especially for the development of oral formulations [16]. However, little research has been reported for dermal or topical delivery, where none include the use of chitosan [17]. In this study, three different prototypes were developed using chitosan (F1), 2:1 chitosan: PVA (F2), and 1:1 chitosan:gelatin (F3). Compatibility tests were done by thermal analysis and Fourier Transformed Infrared spectroscopy (FT-IR). Scanning Electron Microscopy (SEM) was employed to examine the morphology of the surface and inner layers of the scaffolds. Additionally, in vitro release studies were performed at 32 °C and 38 °C to evaluate the release profiles at simulating conditions of skin normothermia and hyperthermia, respectively, which were later adjusted to zero-order, first-order, Hixson–Crowell, Higuchi, and Korsmeyer–Peppas kinetic models for the best formulation.

Therefore, the novelty of this work lies in the development of a suitable topical thermo-responsive chitosan-based scaffold to provide 24 h controlled release of DKT for potential wound healing applications. To our knowledge, no other paper describes such a delivery system that provides a slow release of DKT due to the swelling and dissolution of the polymer matrix, and an increased rate due to its thermo-responsive behavior above the LCST of the system, instead of conventionally loading high drug amounts to provide a concentration gradient for Fickian diffusion as the main release mechanism.

## 2. Materials and Methods

### 2.1. Materials

DKT (99.8% purity) was provided by Laboratorios Infarma LTDA (Cartago, Costa Rica). Chitosan (MW: 100,000–300,000), PVA (MW: 89,000–98,000), and gelatin were purchased from Sigma Aldrich (St. Louis, MO, USA); citric acid monohydrate and benzoic acid from Merck (Kenilworth, NJ, USA). Distilled water was obtained from LABIOFAR facilities (San José, Costa Rica).

### 2.2. Preformulation and Formulation

Literature research was done to obtain information on the potential excipients and manufacturing processes that could be used for developing 25.0 mL units of the topical delivery system to be loaded with 110.9 ± 5.5mg of DKT (equivalent to 75 mg of DK). Special attention was paid to possible physicochemical incompatibilities and technological disadvantages. Therefore, the following prototype formulations (Table 1) were evaluated throughout the study:

Scaffolds were prepared by the solvent casting method. Citric acid was dissolved in distilled water to provide a medium with the desired pH. Then, the solution was heated and maintained at 75 °C for the entire process. DKT was added with continuous agitation until complete dissolution. Benzoic acid, previously dissolved in a small volume of distilled water, was incorporated as well. Chitosan was slowly and steadily added until complete dissolution, avoiding the formation of lumps. For F2 and F3, the secondary polymer was added as done for chitosan. The formed hydrogel was kept with continuous agitation for 2 h and then, subjected to ultrasonication for 1 h to get rid of air bubbles. The formulations were then poured into watch glasses and placed into the oven at 80 °C for 5–6 h to eliminate water. The obtained scaffolds were stored in a desiccator protected from light for 24 h until their use for compatibility tests, SEM analysis, or in vitro release studies.

### 2.3. Compatibility Tests

An assessment of the physicochemical interactions between excipients and the active pharmaceutical ingredient (API) from the prototype formulations was done by analyzing three replicas of each raw material and the formulations presented in Table 1. The analyses were performed under thermal and FT-IR techniques as explained in the following sections.

#### 2.3.1. Thermal Characterization

Thermogravimetric (TG) analyses were performed using a TA Instruments Q500 (New Castle, DE, USA), under an inert nitrogen atmosphere with a 90 mL·min^−1^ sample gas and 10 mL·min^−1^ of balance gas. A sample of approximately 5 mg was added to a platinum crucible and subjected to analysis in the temperature range from 20 to 800 °C with a heating rate of 10 °C min^−1^.

Additionally, Differential Scanning Calorimetry (DSC) was performed using a TA instruments Q200 (New Castle, DE, USA), under an inert nitrogen atmosphere with a 50 mL·min^−1^ gas flow. A sample of approximately 6 mg was added to an aluminum capsule and subjected to an initial isotherm at 20 °C for 5 min, and then to a heating rate of 5 °C min^−1^ in the temperature range from 20 to 350 °C.

Additionally, as proposed by Wang et al., the onset point of the endotherm found within the range from 0 to 100 °C in the DSC thermograms of the formulations, if present, was used to determine the LCST, indicative of the thermo-responsive behavior, which was also taken into account for selecting the best formulation for further analysis [18].

#### 2.3.2. Fourier Transformed Infrared Spectroscopy

Each sample was analyzed using the transmission technique in a Thermo Scientific Nicolet 6700 model (Heredia, Costa Rica) in the range of 500 to 4000 cm^−1^, with 200 scans per sample. Room conditions were maintained at 25 °C and 30% relative humidity.

### 2.4. Microscopic Characterization

The samples of F1, F2, and F3 scaffolds were metalized with a thin layer of gold by sputtering, with an exposure time of 3 min at 20 mA. The analyses were performed by a Jeol JSM-6390LV scanning electron microscope (Peabody, MA, USA), using an acceleration voltage of 5 kV and spot size of 60.

### 2.5. In Vitro Release Studies

The release profile was initially evaluated to 6 units of F1 scaffold prototype, then to 6 units of F2, and finally to F3 units. The study was done according to USP 42 general chapter 〈724〉 Drug Release [19]. The equipment used were Hanson Research Corp Transdermal Sandwich^TM^ devices (Chatsworth, CA, USA), Agilent VARIAN VK 7010 dissolution equipment with automatic sampling (Santa Clara, CA, USA), and apparatus II USP (paddles) at 50 rpm distanced by 25 ± 2 mm from the release device surface. Sink conditions were achieved by using 900.00 mL of distilled water at 32 ± 0.5 °C as the medium and adjusted to pH 6.8 to guarantee API dissolution. A 10.00 mL aliquot from each vessel was taken at 1, 3, 6, 12, 18, and 24 h. A 2.00 mL aliquot from each 10.00 mL sample was taken and diluted in a 10.00 mL volumetric flask. Samples were analyzed using a Thermo Fisher AQUAMATE UV-Vis spectrophotometer (Heredia, Costa Rica) at 260 nm. The scaffold prototype that exhibited: (a) no physicochemical incompatibility, (b) thermo-responsive behavior, (c) the best performance in terms of controlled release, and (d) the major reduction regarding burst release, was selected for further studies to 6 new units following the previously described method, but using the medium at 38 ± 0.5 °C.

### 2.6. Kinetic Release Models

Data from the selected prototype were adjusted to zero-order, first-order, Hixson–Crowell, Higuchi, and Korsmeyer–Peppas kinetic models (Table 2) to determine the main release mechanism and obtain the release rate at 32 °C and 38 °C. The best models were considered to have the highest R^2^ and the lowest Akaike information criterion (AIC), which takes into consideration the residual sum of squares (RSS), the number of parameters (p), and the number of observations (n) as presented in Equation (1):(1)$$\mathrm{AIC}=n\;*\ln\left(\mathrm{SSR}\right)+2p$$

## 3. Results and Discussion

### 3.1. Preformulation and Formulation

Chitosan was chosen as the main polymer of the prototypes due to its outstanding properties for wound healing, such as low toxicity, biocompatibility, biodegradability, and antimicrobial activity [20,21]. The presence of a primary amino group makes the polymer soluble in aqueous solutions of organic acids, such as citric acid at pH 1–4, being the main reason why said acid was used for the formulation process. All developed prototypes in the hydrogel form presented a pH within the intended 3.0–3.5 range to guarantee chitosan dissolution. In addition, the chosen preservative, benzoic acid, exhibits its greatest activity at pH values between 2.5 and 4.5, where the undissociated form is present, which is the one that exerts the antimicrobial activity [22].

However, chitosan incorporation into the formulations represented a challenge in terms of solubility since it requires a different pH compared to the involved API and its concentration. DKT is a weak acid salt, and the pH value of precipitation (pHp) for this substance in the formulations (i.e., the lowest possible pH where the API is still ionized) is 6.12. As a result, a lower pH can cause the precipitation of the API as DK, the lipophilic-weak acid [14]. This technological issue was addressed by heating the solution at 75 °C, which allowed dissolving the DKT at a starting pH of 2.5 at which chitosan is soluble. The initial colloidal solution propitiated the formation of bushy polymeric structures, which combined with chitosan high water absorption, allowed trapping the drug within the macroscopic network, and thus, preventing its precipitation [23].

Despite the great advantages offered by chitosan, the polymer alone is not likely to be suitable for developing controlled release systems [20]. Therefore, the preformulation study allowed identifying two other polymers of interest for developing the topical chitosan-based scaffolds for controlled release: PVA and gelatin. The first one is a synthetic polymer widely known for improving processability and mechanical properties in different applications, whereas gelatin has been extensively used for wound dressings [24,25]. Both polymers are water-soluble and require heating for their dispersion, being compatible with the proposed preparation method. Additionally, they contribute to the biocompatible and biodegradable behavior provided by the chitosan matrix, as well as yield bioadhesive properties to the formulations [24,26].

Figure 1 shows the obtained scaffolds after ultrasonication and dehydration. These prototypes have 7.8 cm of diameter, present an amber color, and a flexible but not elastic consistency. In addition, F2 and F3 scaffolds exhibited preliminary adhesiveness upon contact to test fluid, namely, water, based on the qualitative thumb tack test described by Gutschke et al. [27].

### 3.2. Compatibility Tests

As can be seen in Figure 2a,b, it is not possible to identify the DKT melting point (104.57 °C) in the formulations as obtained in the DSC analysis for the raw material due to its low concentration. In addition, polymerization phenomena due to the agitation and heating time could have caused a series of shifts in different thermal events from the constituent materials [28]. As a result, it is not possible to appreciate chitosan’s endotherm at 90.28 °C or close in any of the formulations. Additionally, all scaffolds show a shift in chitosan exotherm at 305.36 °C to 340–350 °C.

F1 (Figure 2b) shows a crystalline endotherm at 156.45 °C that seems to correspond with the citric acid melting point (154.11 °C). On the other hand, F2 shows a shift in PVA endotherm at 194.30 °C to 176.57 °C, which can be due to the polymerization process and cross-linking with citric acid [28]. However, in this formulation, it is not possible to see PVA second endotherm at 323.61 °C or any other related to chitosan and citric acid. Furthermore, F3 exhibits, at 174.31 °C, a shift from gelatin’s endotherm at 223.81 °C.

Regarding TG (Figure 2c,d), all the formulas exhibited a thermal degradation behavior related to their constituent materials without showing or suggesting any physicochemical incompatibility between them. Likewise, the scaffolds also showed increased thermal stability due to cross-linking [29]. At 219.03 °C, F1 had lost more than 30% of its weight mainly due to the decomposition of citric acid around 210.18 °C. In addition, at 351.41 °C the sample weight was reduced up to 40%, which is explained by the decomposition of more than 50% of the chitosan. F2 shows the same explained degradation behavior, presenting thermal degradation events at 227.45 °C and 331.51 °C. However, for the signal around 406.50 °C, it is important to mention that the sample has only a 36.16% of its initial mass, which is due to a great decomposition of PVA. F3 also showed the mentioned behavior regarding the first two main events, but at 335.34 °C it had only 50.98% of its mass due to gelatin decomposition at 308.56 °C.

Likewise, FT-IR evaluation (Figure 3) supports the obtained results by the thermal analysis. In the near IR region, DK strong bands are seen in the three formulations around 1060 cm^−1^ with vibrations at 1568, 1538, and 771 cm^−1^, which are also indicative of the raw material purity [30]. Chitosan’s O-H stretching close to 3300 cm^−1^ and the C-H vibrations close to 2900 cm^−1^ are observed in the three formulations [31]. Regarding F2, C-H alkyl stretching of PVA can be observed near 2900 cm^−1^, and also, its typical broad band at 3000–3500 cm^−1^. On the other hand, F3 conserves gelatin’s characteristic bands of the amide group close to 3434, 1600, and 1550 cm^−1^, and the asymmetric stretching vibrations are represented with bands between 2950–2800 cm^−1^. However, F2 and F3 showed a decrease in the vibration peak of the O-H stretch (3000–3500 cm^−1^) related to the reduced number of O-H bonds due to cross-linking [32].

### 3.3. Scanning Electron Microscopy Characterization

As can be seen in Figure 4a–c, the surface of the three scaffold samples presents a relatively smooth morphology. However, the F1 scaffold (Figure 4a) showed voids and holes on the surface with diameters in the range of 5 to 10 µm, left by air bubbles possibly due to small variations in the drying process and lower efficiency of the ultrasonication process for that formulation. On the other hand, cross-linked scaffolds 2 and 3 (Figure 4a,b) did not show any evidence of surface holes. Likewise, the addition of PVA (Figure 4e) and gelatin (Figure 4f) in the formulations leads to changes in the morphology of the scaffold observed by the cross-section.

The F1 scaffold (Figure 4d) has an interconnected network and spaces with cave aspects that can exceed 40 µm in length, very different from that observed in the F2 scaffold (Figure 4e) that has a structure like a sponge with the presence of micropores and greater roughness. Furthermore, comparing F1 and F2 with the F3 scaffold (Figure 4f) that contains gelatin, it is possible to identify that this formulation favored the formation of lamellar structures, composed of multiple thin layers arranged in a parallel and compact way. This last morphology could favor a slow release and a reduction of the bursting effect (i.e., an initial large drug bolus released before the stabilization of the release rate) [33].

### 3.4. In Vitro Release Studies

As mentioned before, the literature states a LCST for chitosan around 32 °C, which is considered to be skin normothermia. As a result, a slow drug release should be seen below or at the LCST, depending mainly on surface desorption, swelling, and degradation of the polymer matrix [34]. Figure 5a shows the determination of the LCST for each formulation. As can be seen, F1 shows almost no volume phase transition due to the sole presence of chitosan. The previous is reflected in the release profile of DKT from that formulation (Figure 5b), which exhibited the fastest rate of all prototypes at 32 °C, reaching 88.40 ± 5.79% at 12 h and 102.18 ± 1.39% at 24 h. The controlled release was not achieved with this formulation, instead, burst release is notable from the first hour (31.62 ± 2.17%) until 6 h (75.13 ± 4.18%). Although burst release can be optimal in wound healing for immediate relief, the high burst from this formulation cannot provide the prolonged release that is required for gradual healing, as well as reduces the lifetime of the scaffold [35].

Due to the previous, we evaluated the impact caused by the presence of PVA and gelatin on DKT release from the chitosan-based scaffolds. These polymers were expected to physically cross-link during the manufacturing process to yield a hydrogel, providing the scaffolds with high capability to control the release [36]. Although F2 did not show a greater volume phase transition compared to F1, it provided a reduction in the drug released at 24 h (88.40 ± 3.94%), and also reduced the burst effect (1h: 26.81 ± 3.25%, 6h: 58.16 ± 3.07%). This profile can be attributed to the presence of intra- and inter-cross-linked PVA tubules by the citric acid, which reduces porosity, making the penetration of water moleculesdifficult, and reducing wettability [28].

Nonetheless, F3 was the only formulation that exhibited a significant phase transition (Figure 5a) with an onset temperature at 34.96 °C, which coincides with the cloud point of the system; thus, it is considered as the LCST of this chitosan-based matrix [18]. The prototype showed the best performance in terms of controlled release and reduced the initial burst up to 18.01 ± 0.99% in the first hour and 40.18 ± 1.00% at 6 h. In addition, at 12 h, it delivered a bit more than half of the loaded drug (55.07 ± 2.43%) and a total of 77.75 ± 2.72% at 24 h, approximately equal to 59 mg of DK. The previous release behavior is also related to gelatin physical entrapment and absorption of the API. Initially, gelatin provides a release mostly by diffusion, while the following hours are dominated by its degradation, erosion, and dissolution [37].

Since F3 provided the best performance and exhibited thermo-responsitivity, 6 more units were employed for further studies at 38 °C (Figure 6) to simulate the local hyperthermia conditions caused by the typical inflammatory process presented in wound healing.

It is well known that thermo-responsive systems experience a reversible transition under an entropy-driven process, from a hydrophilic to a hydrophobic state upon a rise in the temperature above the LCST [38]. As a result, the interruption in polymer–water hydrogen bonding, and the increase in hydrophobic interactions within the polymer chains caused the collapse of the structure and the consequent greater burst release at 38 °C (1h: 28.74 ± 2.06%, 6h: 61.05 ± 2.59%) compared to the analyzed units of the same formula at 32 °C [39]. Nevertheless, it is important to notice that this initial burst is still lower than the one exhibited by F1 at 32 °C, which can be attributed to the presence of gelatin as well.

### 3.5. Kinetic Release Models

Rational development of controlled release forms involves in vitro release tests to predict the impact on the in vivo performance of the formulation [40]. R^2^ and AIC are two relevant values that can be used to evaluate how well a set of models fit. The first one quantifies the intensity of the linear relationship between two variables under study, while the second one is a statistical measure for time series models [41].

Table 3 summarizes the most important results from the kinetic release study for F3. For data obtained at 32 °C the fitting to the models was made using all sampled times (n = 6), while at 38 °C only data until 12 h were used (n = 4) because the scaffolds had released more than 85% of the loaded drug by that time, which is enough for making predictions regarding DKT release upon hyperthermia conditions. Regardless of that, Hixson–Crowell and Korsmeyer–Peppas possessed high values for the R^2^ and the lowest values for the AIC at both temperatures for which these can be considered as the best fit models.

The Hixson–Crowell model states that agitation does not affect dissolution rate, which is said to be the controlling factor for the release of the drug instead of its diffusion through the polymer matrix. Although the model considers that dissolution of the matrix causes a surface reduction, it requires the original geometric form of the units to persist [42]. However, Korsmeyer–Peppas seems to be a better model to explain the release profile from the F3 prototype since it has been widely used for describing the release from polymer systems. In addition, it allows explaining the release mechanism where matrix erosion or dissolution takes place [43].

The obtained values for the diffusional exponent (*n*) at 32 °C and 38 °C were respectively 0.457 and 0.450, which represent a non-Fickian transport mechanism and a diffusion-swelling controlled process (*K_KP_* = h^n−1^). The previous is due to the presence of small pores filled with the dissolution medium in the polymer matrix, allowing the subsequent diffusion of DKT through the swelled structure [43]. On the other hand, the first-order model seems to fit only at 32 °C. This model is used for describing drug release from porous structures and it is conditioned by the saturation point of the drug in the medium. Nevertheless, it could not fit the profile at 38 °C since the model is not capable of adjusting the initial data at a higher release rate [42].

## 4. Conclusions

The drug release performance of the developed topical scaffold is influenced by the constituent polymers, chitosan, and gelatin, which allow a sustained release and the reduction of the initial burst. In addition, natural feedback from the host expressed as local hyperthermia over the LCST of the polymer matrix can modify the release of the loaded drug as well. As a result, this scaffold can be used to deliver the necessary dose according to each patient in order to provide the therapeutic effect; thus, it can be considered as a suitable approach for wound healing with a controlled release of DKT for 24 h use that can also potentially reduce side effects, allowing overcoming adherence issues and wound healing complications. However, further studies are recommended to design matrix and reservoir topical thermo-responsive systems that provide controlled release for more than 24 h. It is also relevant to study the release profile of the drug considering other aspects like the wound pH presented throughout the different stages of the healing process. Furthermore, future research should perform in vivo studies for cell proliferation and anti-inflammatory effect evaluation, as well as assess the allergenic potential that may affect sensitized individuals to seafood due to chitosan synthesis from crustacean shells.

## Figures and Tables

**Figure 1 pharmaceutics-13-02100-f001:**
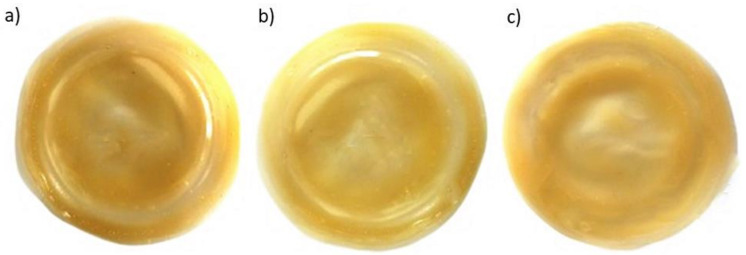
Thermo-responsive scaffold prototypes for DKT controlled release: (**a**) F1; (**b**) F2; (**c**) F3.

**Figure 2 pharmaceutics-13-02100-f002:**
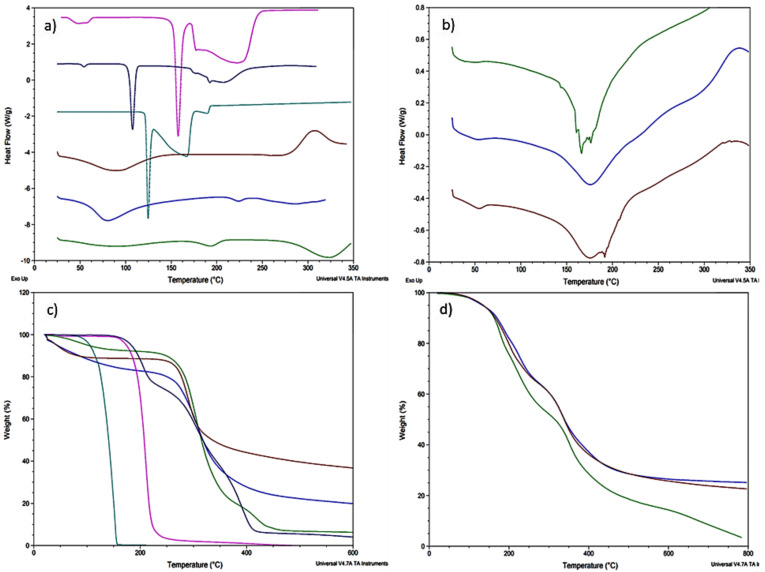
Thermal analysis; (**a**) DSC analysis of raw materials: chitosan (wine), PVA (green), gelatin (blue), DKT (dark blue), citric acid (fuchsia), and benzoic acid (aqua); (**b**) DSC analysis of the formulation prototypes: F1 (green), F2 (blue), and F3 (wine); (**c**) TG analysis of raw materials: chitosan (wine), PVA (green), gelatin (blue), DKT (dark blue), citric acid (fuchsia), and benzoic acid (aqua); (**d**) TG analysis of the formulation prototypes: F1 (green), F2 (blue), and F3 (wine).

**Figure 3 pharmaceutics-13-02100-f003:**
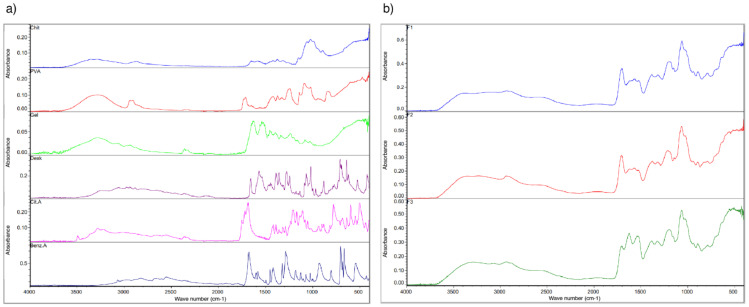
FT-IR spectroscopic analysis; (**a**) raw materials: chitosan (blue), PVA (red), gelatin (green), DKT (purple), citric acid (fuchsia), and benzoic acid (blue); (**b**) formulation prototypes: F1 (blue), F2 (red), and F3 (green).

**Figure 4 pharmaceutics-13-02100-f004:**
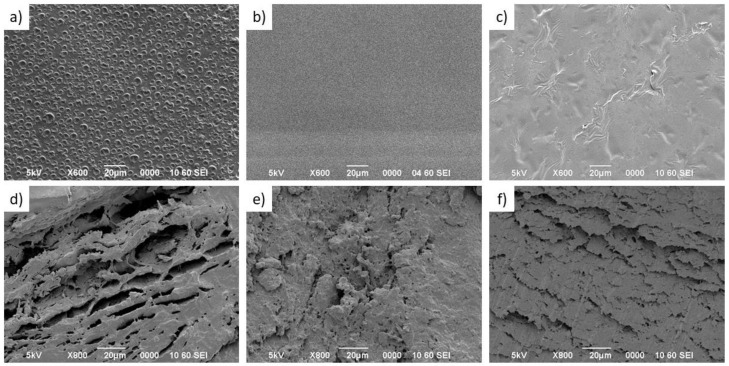
SEM micrographs of the surface (**a**–**c**) and cross sections (**d**–**f**) of three formulated scaffolds (scale bar: 20 μm); (**a**,**d**) F1 scaffold, (**b**,**e**) F2 scaffold, and (**c**,**f**) F3 scaffold.

**Figure 5 pharmaceutics-13-02100-f005:**
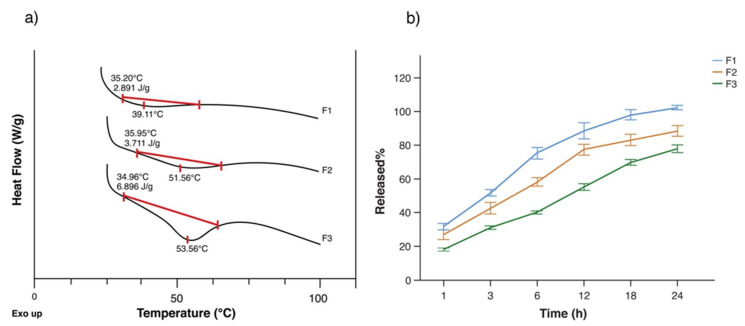
Drug release: (**a**) determination of LCST; (**b**) average drug release comparison between the three prototypes at 32 °C: F1 (blue), F2 (red), and F3 (green) (data are presented as mean of ± standard deviation, n = 6).

**Figure 6 pharmaceutics-13-02100-f006:**
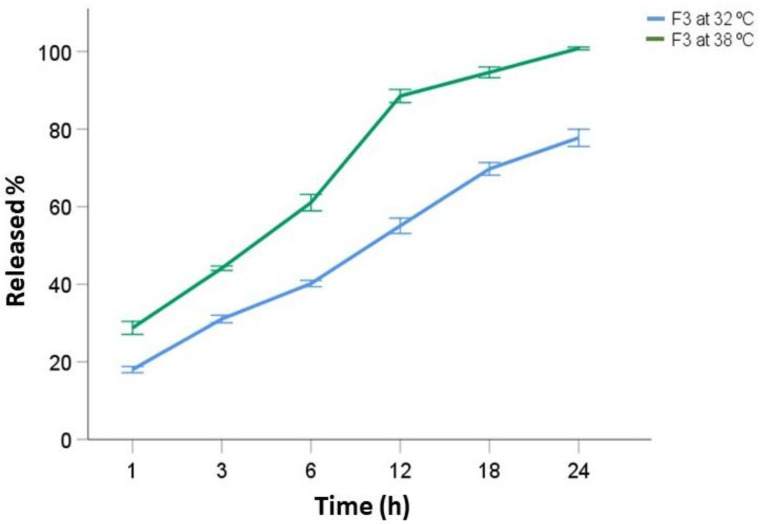
F3 average drug release comparison at 32 °C (blue) and 38 °C (green) (data are presented as mean of ± standard deviation, n = 6).

**Table 1 pharmaceutics-13-02100-t001:** Scaffolds prototypes for DKT controlled release.

Composition (*w*/*v* %)	F1	F2	F3
DKT	0.44	0.44	0.44
Chitosan	4	4	4
PVA	-	2	-
Gelatin	-	-	4
Citric acid	5	5	5
Benzoic acid	0.2	0.2	0.2
Distilled water	q.s 25.0 mL	q.s 25.0 mL	q.s 25.0 mL

DKT: Dexketoprofen trometamol, PVA: Polyvinyl alcohol, F1: Formulation 1, F2: Formulation 2, F3: Formulation 3.

**Table 2 pharmaceutics-13-02100-t002:** Kinetic models for drug release.

Model	Equation		Terms
Zero-order	$Q_{t}=K_{0}\;*\;t$	(2)	$Q_{t}$: amount of drug dissolved in time t$;\;K_{0}$: zero-order release constant in mg·h^−1^
First-order	$\frac{dC}{dt}=-K\;*\;C$	(3)	$\frac{dC}{dt}$: change in concentration C with respect to change on time; K: first-order release constant in h^−1^
Hixson–Crowell	$W_{0^{1/3}}-W_{t^{1/3}}=K_{\mathrm{HC}}\;*\;t$	(4)	$W_{0}$$:\;\mathrm{initial}\;\mathrm{amount}\;\mathrm{of}\;\mathrm{drug};\;W_{t}$: remaining amount of drug in the dosage form at time t$;\;K_{HC}$: Hixson–Crowell constant in mg^1/3^ h^−1^
Higuchi	$Q=K_{H}\sqrt{t}$	(5)	Q$:\;\mathrm{cumulative}\;\mathrm{quantity}\;\mathrm{of}\;\mathrm{drug}\;\mathrm{released};\;K_{H}$: Higuchi constant in h^0.5^
Korsmeyer–Peppas	$\frac{M_{t}}{M_{\infty }}=K_{\mathrm{KP}}{\;*\;t}^{n}$	(6)	$\frac{M_{t}}{M_{\infty }}$*:* fraction of drug released at a certain time t$;\;n:\;\mathrm{diffusional}\;\mathrm{exponent};\;K_{KP}$$:\;\mathrm{Korsmeyer}\;\mathrm{release}\;\mathrm{rate}\;\mathrm{constant}\;\mathrm{in}\;t^{-0.5}$$\;\mathrm{for}\;\mathrm{Fickian}\;\mathrm{diffusion},\;\mathrm{or}\;\mathrm{in}\;t^{n-1}$ for non-Fickian diffusion

**Table 3 pharmaceutics-13-02100-t003:** Kinetic release models for F3 at 32 °C during 24 h and at 38 °C during 12 h.

Model	32 °C			38 °C		
	Release Rate	R^2^	AIC	Release Rate	R^2^	AIC
Zero-order	2.70 ± 0.25	0.9661	29.81	5.70 ± 0.43	0.9886	15.00
First-order	0.0557 ± 0.0016	0.9968	−29.01	0.167 ± 0.016	0.9827	−9.53
Hixson–Crowell	0.0669 ± 0.0028	0.9931	−24.18	0.1759 ± 0.0074	0.9965	−17.52
Higuchi	16.51 ± 0.41	0.9976	14.08	26.1 ± 1.1	0.9965	10.34
Korsmeyer–Peppas	0.181 ± 0.023	0.9979	−32.48	0.279 ± 0.040	0.9941	−20.06

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
