# Peer review of "Topical Chitosan-Based Thermo-Responsive Scaffold Provides Dexketoprofen Trometamol Controlled Release for 24 h Use"

_pharmaceutics, 2021, doi:10.3390/pharmaceutics13122100_

Round 1

Reviewer 1 Report

This research report summarizes the research related to the development of dexketoprofen chitosan scaffolds for cotrolled release.

  1. The title of the paper suggests that the proposed formulation is for wound healing, however the autors did not show any proof of concept for application of the synthesized material for wound healing. Also the authors failed to explain the adantages of a thermo responsie release in wound healing.
  2. Similar materials have been reported with same drug for other applications. For example; Microvasc Res
    . 2020 Mar;128:103961. doi: 10.1016/j.mvr.2019.103961. Epub 2019 Nov 20."Treatment of oxidative stress-induced pain and inflammation with dexketoprofen trometamol loaded different molecular weight chitosan nanoparticles: Formulation, characterization and anti-inflammatory activity by using in vivo HET-CAM assay"..Hence I feel that in the current form, the work lacks significant scientific merit and does not make any noticeable improvement to the intended field of research. I sugges the authors to consider adding proof of concept by including some in vivo wound healing data or anti-inflammtory or analgesic action.

Author Response

REVIEWER #1

Dear reviewer,

Our research group is highly grateful to you for your comments and suggestions for improving the manuscript. We made different changes throughout the text, as well as revised English writing. Here, we respectfully give answer to your different observations:

“The title of the paper suggests that the proposed formulation is for wound healing, however the autors did not show any proof of concept for application of the synthesized material for wound healing. Also the authors failed to explain the advantages of a thermo responsive release in wound healing.”

We changed the title of the paper to not address directly the wound healing application as the highlight of the research, but to focus the relevance of our work in the developed delivery system, its physicochemical and kinetic characterization, as well as the exhibited controlled release performance. As a result of that and according to our findings, we suggest that the developed system possess a potential to be used for wound healing based on the controlled release at normothermia conditions and upon a thermal feedback, such as presented in inflammatory process in wound healing. To prove the previous, we modified figure 5 (page 9) to present the LCST determination of each formulation and the release profile at 32°C. It is possible to observe that only one scaffold (F3) showed thermo-responsitivity, which was actually the same one that exhibited the best controlled release and the most reduced burst effect. In addition, the advantages of this controlled released influenced by the thermo-responsive behavior of the matrix are mentioned in the manuscript, such as reducing dosing, patient compliance and a release that is more personalized, according to each individuals response in wound healing.

“Similar materials have been reported with same drug for other applications. For example; Microvasc Res. 2020 Mar;128:103961. doi: 10.1016/j.mvr.2019.103961. Epub 2019 Nov 20."Treatment of oxidative stress-induced pain and inflammation with dexketoprofen trometamol loaded different molecular weight chitosan nanoparticles: Formulation, characterization and anti-inflammatory activity by using in vivo HET-CAM assay". Hence I feel that in the current form, the work lacks significant scientific merit and does not make any noticeable improvement to the intended field of research. I suggest the authors to consider adding proof of concept by including some in vivo wound healing data or anti-inflammatory or analgesic action.”

The research that you mentioned was cited by us in the introduction (reference 16), where we stated that effectively, there has been some approaches with chitosan and DKT for controlled delivery, but they have not been employed for the topical administration route. Actually, that research involved the oral route, which has been the most studied one for DKT delivery. Therefore, our approach implies an improvement, since we are not looking for a systemic effect but a local one, thus potentially reducing the side effects. In addition, rational development of drug delivery systems for any drug intended to be administered in any route requires an initial screening by in vitro studies, which have demonstrated good correlation with the in vivo performance in many cases. As mentioned before, the main focus of this paper was to develop a system that provides a controlled release during 24h and that is responsive to thermal stimulus, not by Brownian diffusion but by the phase transition of the polymer, which allows releasing more material upon an increase in temperature over the LCST. Therefore, our goal was to characterize the system in terms of biopharmaceutical aspects, physicochemical compatibility, in vitro dissolution performance, and kinetic release, not in vivo evaluation.

Reviewer 2 Report

The authors aim at developing a novel controlled release systems composed of a chitosan-based scaffold for transdermal delivery of COX inhibitor ̶-Dexketoprofen trometamol (DKT). Starting from the cross-linking and loading process, a technical problem regarding dissolution and mixing of materials like chitosan and DKT was overcome. Two polymers PVA and gelatin were employed in the scaffold to form different formulations, too and were analytically examined regarding their compatibility with the loading of DKT through analytical methods, using DSC, TGA and FTIR. The formulations showed also distinct structures that might trigger the release profiles. The systems particularly involving gelatin could efficiently reduce the burst release of DKT during the first few hours as the release was sustained over 24 h.  Furthermore, the burst release in the system could be controlled under hyperthermia condition that may potentially reduce side effect during medication and overcome wound healing complications. Although authors made the hypothesis based on corresponding results in a logical manner, there is some lack of explanations and inaccurate descriptions in the study. I have some concerns regarding several issues in the following comments that should be discussed and addressed before the work can be published

  1. The introduction about the thermoresponsivity of chitosan provided by the reference should be more precisely described as thermoresponsive properties do not exist in unmodified chitosan. Typically, it can only exhibit thermoresponsive behavior when it is modified or cross-kinked with other materials (e.g. glycerophosphate, PNIPAM). A self-citation with such claim is not satisfying (page 2, line 55-58, section 1)
  2. The advantages of the materials used in this study should be explained in more detail (e.g. what is the role of citric acid in the system?). Among many polymers that can physically or chemically cross-linked with chitosan, why did authors decide to use PVA and gelatin in the chitosan-base scaffolds for drug loading and control release? Is it related to the compatibility with DKT or thermoresponsivity?
  3. Why did the authors adjust medium pH value to 6.8 for testing the release profile? (page 4, line 143, section 2.5)
  4. Regarding PVA and gelatin incorporation into the scaffolds; what did the test fluid used for adhesiveness? Was there any difference in the degree of adhesiveness among the three formulations? (page 5, line 184-185, section 3.1)
  5. According to the results derived from DSC ad TG analysis, the formulations showed some signals corresponding to the raw materials. However, the proof of thermoresponsivity is not provided by these two analysis methods, as LCST of chitosan should be at around 32 °C claimed by authors. Generally, when measuring a LCST of polymers like PNIPAM where phase separation occurs at 32 °C, a signal can be observed at 32 °C due to loss of hydrogen bonds leading to a collapse of conformation and therefore representing an endothermic process. None of these three formulations showed this sign. Thus, the following comments should be carefully considered:

1) What is the mechanism of different formulations formed with PVA and gelatin and loaded with DKT at 75 °C? Since DKT release profile for the formulation 3 at 32 and 38°C fit best in Korsmeyer-Peppas model, what is the difference of DKT released from the formulation 3 at 32 and 38 °C abide by this kinetic model?  Authors have explained how DKT diffuse at 32 °C, but the explanation for 38 °C is not clear. (page 10, line 323-331, section 3.5)

2) The released amount of DKT was calculated over a period of time measured by UV-vis spectrometer. However, how did the authors calculate the loading amount and determine the release percentage in the three formulations? The loading amount can be different based on different formulations, structures, and porosity, in other words, higher loading capacity may release more compared to lower one with a delay of release. In Figure 5 a), the release profiles of all formulation were similar, were the loading amounts assumed to be the same?

3) At 32 °C, it is considered that DKT diffuses out through the swollen structure, while the collapsed structure due to decreasing hydrogen bonds and rising hydrophobic interaction cause a burst release at 38 °C. In such case, there is no solid support found in this study to demonstrate the conformational change, structure change, and the swelling degree of the formulations. Why did the authors claim the systems are thermoresponsive? Typically, an increasing temperature is followed by the rising diffusion rate, attributed to thermal energy gained. Isa it possible that the faster release at 38 °C in formulation 3 follow this aspect? (page 9, line 285-295, section 3.4)

Author Response

REVIEWER #2

Dear reviewer,

Our research group is highly grateful to you for your comments and suggestions for improving the manuscript. We made different changes throughout the text, as well as revised English writing. Here, we respectfully give answer to your different observations:

“The introduction about the thermoresponsivity of chitosan provided by the reference should be more precisely described as thermoresponsive properties do not exist in unmodified chitosan. Typically, it can only exhibit thermoresponsive behavior when it is modified or cross-linked with other materials (e.g. glycerophosphate, PNIPAM). A self-citation with such claim is not satisfying (page 2, line 55-58, section 1)”

We made the clarification that the thermo-responsive property can be induced by proton transfer from the material to another cooperatively binded compound (e.g., β-glycerophosphate), or by the cross-linking with another polymer, and added more supporting references to that fact (see paragraph 1, page 2 of the new manuscript version).

“The advantages of the materials used in this study should be explained in more detail (e.g. what is the role of citric acid in the system?). Among many polymers that can physically or chemically cross-linked with chitosan, why did authors decide to use PVA and gelatin in the chitosan-base scaffolds for drug loading and control release? Is it related to the compatibility with DKT or thermoresponsivity?”

We addressed the contribution of PVA and gelatin briefly in the introduction (see paragraph 1, page 2 of the new manuscript version), however, a stronger description of their advantages are discussed in page 5, third paragraph, where we basically support the choice of these materials based on their possible contribution to the biocompatible and biodegradable behavior, the suitability for the manufacturing process of the scaffolds (water-solubility, heating), their preliminary compatibility with the other excipients to be introduced in the formulation, as well as their capacity for controlled release. Regarding the other materials, in page 5 the first paragraph states the relevance of citric acid for adjusting the pH to adequate values for chitosan solubility, as well as for the antimicrobial activity of the added preservative, benzoic acid.

“Why did the authors adjust medium pH value to 6.8 for testing the release profile? (page 4, line 143, section 2.5)”

The medium was adjusted properly to study the correct dissolution (included in the methodology) of the API and having a controlled pH throughout the different tests at 32°C and 38°C for the three formulations, therefore minimizing the influence of that variable. In addition, skin pH is usually acidic (5.5), but in wounds pH is disrupted and gets closer to a neutral value. Although this paper is not about the influence of the pH over this kind of extracellular matrix we wanted to work closer to neutrality since the API precipitates below 6.12.

“Regarding PVA and gelatin incorporation into the scaffolds; what did the test fluid used for adhesiveness? Was there any difference in the degree of adhesiveness among the three formulations? (page 5, line 184-185, section 3.1)”

The test fluid was water, which was used for slightly moisturizing the surface of the scaffold. We employed the qualitative tests for assessing preliminary adhesiveness, named, the thumb tack test (see page 5, last paragraph). In this, the thumb is slightly put into contact with the sample for a short time and then, quickly withdrawn. We varied the applied pressure and the time, and noted in all cases for F2 and F3 that a strong bond was formed between the matrix and the skin. By slightly moisturizing the scaffolds surface we simulated the hydrophilic wound environment, which favors the adhesion of the scaffolds due to their hydrophilic nature too. However, due to the subjectivity nature of this test, we cannot make a strong comparison between F2 and F3, but we can say that both exhibited adequate adhesiveness when we did the test at 180° and 90°. In contrast, F1 exhibited some sort of adhesiveness but not as good compared to F2 and F3. Therefore, we must say that gelatin not only allows controlling the release of the API, but also provides the formulation with adhesive properties, since it is also considered a pressure-sensitive adhesive. On the other hand, PVA also has some excellent film- forming and adhesive properties.

“According to the results derived from DSC ad TG analysis, the formulations showed some signals corresponding to the raw materials. However, the proof of thermoresponsivity is not provided by these two analysis methods, as LCST of chitosan should be at around 32 °C claimed by authors. Generally, when measuring a LCST of polymers like PNIPAM where phase separation occurs at 32 °C, a signal can be observed at 32 °C due to loss of hydrogen bonds leading to a collapse of conformation and therefore representing an endothermic process. None of these three formulations showed this sign.”

We would like to apologize for not providing such an important result for our research. To prove the previous, we modified figure 5 (page 9) to present the LCST determination of each formulation and the release profile at 32°C. It is possible to observe that only one scaffold (F3) showed thermo-responsitivity, with an onset temperature at 34.96°C for the endotherm found within the temperature range from 0-100°C. A more descriptive explanation can be found along section 3.4. In vitro release studies. In addition, we included the determination of LCST in the methodology as part of the thermal characterization (page 3, sub section 2.3.1). Since the provided figure in the paper suffered some edition process for including the three scaffolds, here we are presenting to you the original thermogram of each, where you can confirm the data provided in the paper, but specially, the onset of the phase transition suffered by the polymer in F3:

F1

F2

F3

“What is the mechanism of different formulations formed with PVA and gelatin and loaded with DKT at 75 °C? Since DKT release profile for the formulation 3 at 32 and 38°C fit best in Korsmeyer-Peppas model, what is the difference of DKT released from the formulation 3 at 32 and 38 °C abide by this kinetic model?  Authors have explained how DKT diffuse at 32 °C, but the explanation for 38 °C is not clear. (page 10, line 323-331, section 3.5)”

The aforementioned also supports the kinetic results obtained at 32°C and 38°C for the formulation, where the release at 38°C is not merely by the Brownian diffusion or thermal energy gained, but by the phase transition, which causes the collapse of the structure.

“The released amount of DKT was calculated over a period of time measured by UV-vis spectrometer. However, how did the authors calculate the loading amount and determine the release percentage in the three formulations? The loading amount can be different based on different formulations, structures, and porosity, in other words, higher loading capacity may release more compared to lower one with a delay of release. In Figure 5 a), the release profiles of all formulation were similar, were the loading amounts assumed to be the same?”

As newly mentioned in the introduction, part of the novelty of the work is based on the fact that our delivery system does not need to be loaded with high amounts of API to provide a concentration gradient for Fickian diffusion, but the release is from the beginning controlled by a non-fickian diffusion due to the swelling and dissolution of the polymer matrix, and above the LCST (34.96 °C for F3), by the collapsing structure due to the phase transition. All the scaffolds were loaded with almost the same amount of drug, which is actually the one to be delivered during 24 hours. Here, we show you the loaded drug in each case, which was used for calculating the released percentage:

mg loaded / unit

Prototype

#1

#2

#3

#4

#5

#6

1

108,3

109,2

108,9

109,3

109,2

109,4

2

106,5

106,3

105,9

107,4

107,9

106,9

3

106,3

107,8

108,3

107,7

108,4

107,7

All formulations were initially loaded with approximately 111 mg of DKT. The beaker (empty weight known) was weighted after ultrasonification (before casting) and after casting.  Knowing the exact amount of API weighted in each case and the residue in each beaker allowed us to know how much API was actually incorporated into each unit. All measurements were done in an analytical scale.

“At 32 °C, it is considered that DKT diffuses out through the swollen structure, while the collapsed structure due to decreasing hydrogen bonds and rising hydrophobic interaction cause a burst release at 38 °C. In such case, there is no solid support found in this study to demonstrate the conformational change, structure change, and the swelling degree of the formulations. Why did the authors claim the systems are thermoresponsive? Typically, an increasing temperature is followed by the rising diffusion rate, attributed to thermal energy gained. Isa it possible that the faster release at 38 °C in formulation 3 follow this aspect? (page 9, line 285-295, section 3.4)”

This question is answered by previous explanations.

Reviewer 3 Report

Major comments:

Introduction: the introduction needs to be rewritten to highlight novelty aspects and significance of this research.

The authors claim a thermoresponsiveness for chitosan without showing in the introduction nor the results section type of their responsiveness and rheological characteristics at different temperature 32 nd 38oC.

I do not agree with the authors to descirbe trandermal delivery to the formulation in question. The intended effect was to treat local wound; therefore, a topical chitosan-based scaffold is more appropriate.

casting the formulation in a glass watch (round bottom) does not allow uniform distribution of drug throughout the polymeric scaffold and this is very clear from the unequal distribution of the polymeric material and color. dark brown in the centre and lighter brown/yellow colour at the perimeter.

Why not ketoprofen (the commercially available drug) instead of dexketoprofen used in this study?

the discussion is very poor. The role of each excipient used in the formulation was not clearly explained.

Redundant information like details of the calibration curve and equation of Beer's law should be removed. 

Kinetic equation should be in the method section.

In vivo evaluation must be provided for publication in this high-impact  journal.

Author Response

REVIEWER #3

Dear reviewer,

Our research group is highly grateful to you for your comments and suggestions for improving the manuscript. We made different changes throughout the text, as well as revised English writing. Here, we respectfully give answer to your different observations:

“Introduction: the introduction needs to be rewritten to highlight novelty aspects and significance of this research.”

We rewrote the introduction making emphasis on the novelty and highlights of the research. The novelty lies in the development of a suitable topical thermo-responsive chitosan-based scaffold to provide 24-hour controlled release of DKT for potential wound healing applications. To our knowledge, no other paper describes such a delivery system that provides slow release of DKT due to the swelling and dissolution of the polymer matrix, and an increased rate due to its thermo-responsive behavior above the LCST of the system, instead of conventionally loading high drug amounts to provide a concentration gradient for Fickian diffusion as the main release mechanism.

“The authors claim a thermoresponsiveness for chitosan without showing in the introduction nor the results section type of their responsiveness and rheological characteristics at different temperature 32 and 38°C.”

We would like to apologize for not providing such an important result for our research. To prove the previous, we modified figure 5 (page 9) to present the LCST determination of each formulation and the release profile at 32°C. It is possible to observe that only one scaffold (F3) showed thermo-responsitivity, with an onset temperature at 34.96°C for the endotherm found within the temperature range from 0-100°C. A more descriptive explanation can be found along section 3.4. In vitro release studies. In addition, we included the determination of LCST in the methodology as part of the thermal characterization (page 3, sub section 2.3.1). Since the provided figure in the paper suffered some edition process for including the three scaffolds, here we are presenting to you the original thermogram of each, where you can confirm the data provided in the paper, but specially, the onset of the phase transition suffered by the polymer in F3:

F1 see attached the file

F2 see attached the file

F3 see attached the file

The aforementioned also supports the kinetic results obtained at 32°C and 38°C for the formulation, where the release at 38°C is not merely by the Brownian diffusion or thermal energy gained, but by the phase transition, which causes the collapse of the structure. As newly mentioned in the introduction, part of the novelty of the work is based on the fact that our delivery system does not need to be loaded with high amounts of API to provide a concentration gradient for Fickian diffusion, but the release is from the beginning controlled by a non-fickian diffusion due to the swelling and dissolution of the polymer matrix, and above the LCST (34.96 °C for F3), by the collapsing structure due to the phase transition.

“I do not agree with the authors to describe transdermal delivery to the formulation in question. The intended effect was to treat local wound; therefore, a topical chitosan-based scaffold is more appropriate.”

We accept the modification regarding the application, which indeed is intended to be local, therefore, we modify everything to be “topical” or “dermal” in the text.

“Casting the formulation in a glass watch (round bottom) does not allow uniform distribution of drug throughout the polymeric scaffold and this is very clear from the unequal distribution of the polymeric material and color. dark brown in the centre and lighter brown/yellow colour at the perimeter.”

Regarding the manufacturing, actually casting into a glass watch shouldn’t be related with distribution issues, since the solution was homogeneous, but the color changes you mention are due to the drying process since the centre has a slightly thicker layer. In addition, the studied systems present a release due to the erosion (dissolution) of the matrix without losing its form, and different other local sustained release dosage forms have this concave forms, such as loramyc, where actually this could be beneficial for better adherence due the irregular surface of wounds.

“Why not ketoprofen (the commercially available drug) instead of dexketoprofen used in this study?”

We choose dexketoprofen instead of ketoprofen because it is a more novel molecule which is more potent, with better profile in terms of side-effects, and it is also commercially available in tablets, oral solutions and gels. We include part of this in the introduction (see page 2, paragraph 2).

“The role of each excipient used in the formulation was not clearly explained.”

We addressed the contribution of PVA and gelatin briefly in the introduction (see paragraph 1, page 2 of the new manuscript version), however, a stronger description of their advantages are discussed in page 5, third paragraph, where we basically support the choice of these materials based on their possible contribution to the biocompatible and biodegradable behavior, the suitability for the manufacturing process of the scaffolds (water-solubility, heating), their preliminary compatibility with the other excipients to be introduced in the formulation, as well as their capacity for controlled release. Regarding the other materials, in page 5 the first paragraph states the relevance of citric acid for adjusting the pH to adequate values for chitosan solubility, as well as for the antimicrobial activity of the added preservative, benzoic acid.

“Redundant information like details of the calibration curve and equation of Beer's law should be removed. Kinetic equation should be in the method section.”

We removed basic and redundant information from the introduction and the methodology, and added the kinetic equations in the methodology.

“In vivo evaluation must be provided for publication in this high-impact journal.”

Our approach implies an improvement, since we are not looking for a systemic effect but a local one, thus potentially reducing the side effects. In addition, rational development of drug delivery systems for any drug intended to be administered in any route requires an initial screening by in vitro studies, which have demonstrated good correlation with the in vivo performance in many cases. As mentioned before, the main focus of this paper was to develop a system that provides a controlled release during 24h and that is responsive to thermal stimulus, not by Brownian diffusion but by the phase transition of the polymer, which allows releasing more material upon an increase in temperature over the LCST. Therefore, our goal was to characterize the system in terms of biopharmaceutical aspects, physicochemical compatibility, in vitro dissolution performance, and kinetic release, not in vivo evaluation.

Reviewer 4 Report

Include some recent references to the discussion part.

what could be the future scope for the work.

Author Response

REVIEWER #4

Dear reviewer,

Our research group is highly grateful to you for your comments and suggestions for improving the manuscript. We made different changes throughout the text, as well as revised English writing. Here, we respectfully give answer to your different observations:

“Include some recent references to the discussion part”

We included more recent references in the discussion part as well as new ones derived from aspects that were not discussed in the original manuscript, like the role of PVA, gelatin, citric acid and benzoic acid, the demonstration of the LCST, and the support for the thermal behavior of the formulations. In addition, we also added some relevant new information and references in the introduction for highlighting the novelty of the developed delivery system.

“What could be the future scope for the work.”

Regarding this, we added the following statement in the conclusion:

“However, further studies are recommended to design matrix and reservoir topical thermo-responsive systems that provide controlled release for more than 24 hours. It is also relevant to study the release profile of the drug considering other aspects like the wound pH presented throughout the different stages of the healing process. Furthermore, future research should perform in vivo studies for cell proliferation and anti-inflammatory effect evaluation, as well as assess the allergenic potential that may affect sensitized individuals to seafood due to chitosan synthesis from crustacean shells. “

Round 2

Reviewer 1 Report

I have no more comments. The reviewers addressed my concerns.